# The Discovery and Function of Filaggrin

**DOI:** 10.3390/ijms23031455

**Published:** 2022-01-27

**Authors:** J. Kenneth Hoober, Laura L. Eggink

**Affiliations:** Susavion Biosciences, Inc., 1615 W University Drive, Suite 132, Tempe, AZ 85281, USA; Laura.eggink@susavion.com

**Keywords:** keratohyalin granules, histidine-rich protein, filaggrin, profilaggrin, loss-of-function mutations, ichthyosis vulgaris, atopic dermatitis, eczema, corneodesmosomes, transglutaminase

## Abstract

Keratohyalin granules were discovered in the mid-19th century in cells that terminally differentiate to form the outer, cornified layer of the epidermis. The first indications of the composition of these structures emerged in the 1960s from a histochemical stain for histidine, followed by radioautographic evidence of a high incidence of histidine incorporation into newly synthesized proteins in cells containing the granules. Research during the next three decades revealed the structure and function of a major protein in these granules, which was initially called the ‘histidine-rich protein’. Steinert and Dale named the protein ‘filaggrin’ in 1981 because of its ability to aggregate keratin intermediate filaments. The human gene for the precursor, ‘profilaggrin,’ was reported in 1991 to encode 10, 11 or 12 nearly identical repeats. Remarkably, the mouse and rat genes encode up to 20 repeats. The lifetime of filaggrin is the time required for keratinocytes in the granular layer to move into the inner cornified layer. During this transition, filaggrin facilitates the collapse of corneocytes into ‘building blocks’ that become an impermeable surface barrier. The subsequent degradation of filaggrin is as remarkable as its synthesis, and the end-products aid in maintaining moisture in the cornified layer. It was apparent that ichthyosis vulgaris and atopic dermatitis were associated with the absence of this protein. McLean’s team in 2006 identified the cause of these diseases by discovering loss-of-function mutations in the profilaggrin gene, which led to dysfunction of the surface barrier. This story illustrates the complexity in maintaining a healthy, functional epidermis.

## 1. Keratohyalin Granules and Histidine

Karen Holbrook [1] commented that “morphology is often the starting point of an investigation”. Developments in microscopy during the 19th century opened morphology of the cellular world to full view. The ability to thin-section fixed tissues and innovations in selectively staining cellular components provided biologists with opportunities to study structures of tissues and the organelles within cells. This brief historical review starts with several striking observations made by histochemical and radioautographic analyses of the epidermis, which, as the story is told, led to the discovery of filaggrin and its short-lived but essential functions in the assembly of a healthy surface barrier.

A fascinating morphological feature forms under the cornified layer of the epidermis (Figure 1) as keratinocytes move outward from the proliferative basal layer and terminally differentiate into corneocytes that form the surface barrier of the skin. Stephen Rothman [2], recounting the early history of studies on these ‘keratohyalin granules’, remarked that during those slower years of “wax candles and horse carriages”, there was time for “detailed and precise observations that revealed the existence of granules in the granular layer, [which were] first observed and recorded by Auffhammer (1869)”. The granules were further studied by Langerhans (1873) and designated ‘keratohyalin’ in 1882 by Waldeyer. Although very active in research on the skin throughout his career, Rothman commented shortly before he died in 1963 that not much more had been learned since those early days [3]. He did not live to see the dramatic developments that emerged from the investigations of these unusual structures that were already underway in several laboratories.

Seminal observations were published in the years following 1960. Reaven and Cox [5,6] traced the accumulation of zinc in the granular layer to chelation by the high density of histidine in the keratohyalin granules [7]. The granules stained an intense red color with diazotized sulfanilic acid under alkaline conditions, a coupling reaction developed by Pauly [8] for histidine (Figure 2A). Kimie Fukuyama, while a visiting scientist in I. A. Bernstein’s laboratory at the University of Michigan, began investigations of the epidermis by the incorporation of radiolabeled nucleotides into DNA and then into RNA by radioautography. She observed that, whereas DNA was synthesized only in the basal layer, incorporation of precursors into RNA occurred throughout the layers below the stratum corneum [9,10], which suggested that proteins were also synthesized in the outer layers. The surprise came when she turned to the incorporation of [^3^H] amino acids. As expected, labeled phenylalanine, leucine and methionine were extensively incorporated into the dividing cells of the basal layer but minimally in the outer layers. Conversely, [^3^H]histidine and [^3^H]glycine were preferentially incorporated into the granular layer (Figure 2B) [11,12,13]. The possibility of synthesis of a unique protein exclusively in the granular layer while the cells are undergoing terminal differentiation was proposed by Bernstein, who had become interested in the epidermis as a tissue to study the process of differentiation [14]. This hypothesis called for a deeper analysis, which could only be resolved by purification of a protein enriched in histidine and glycine.

## 2. Discovery of the ‘Histidine-Rich’ Protein

This goal was accomplished in Bernstein’s laboratory by one of the authors (JKH) while a graduate student. A protein fraction was extracted from isolated epidermis of newborn rats that contained several-fold more histidine, and a five- to six-fold greater specific radioactivity from injected [^3^H]histidine or [^3^H]glycine, than the remaining bulk protein [15,16]. Subsequent graduate students and postdoctoral fellows continued to refine the study of this protein fraction, which became known as the ‘histidine-rich protein (HRP)’ and was definitively localized to the granular layer by dissociation of the lower layers of the epidermis with tetraphenylboron [17]. Extraction of the granular and cornified layers with 8 M urea, dialysis, lyophilization and extraction of the dried material with 0.1 N perchloric acid followed by precipitation at pH 4.5 (the protocol developed by Hoober [4]) yielded a basic protein fraction that contained 7% histidine, 12% arginine, 16% serine and 14% glycine [16,17,18]. The puzzling observation that 8 M urea was required for extraction of a protein inherently water soluble because of the high content of polar amino acids would become clear many years later. Ball et al. [19] prepared a high molecular weight HRP from the granular layer that was converted to a smaller form in the stratum corneum, with kinetics of an apparent precursor/product relationship. Ugel [20,21] extracted proteins from bovine hoof epidermis with 1 M potassium phosphate, pH 7.0, which when dialyzed formed aggregates similar to keratohyalin granules. Application of this technique to newborn rat epidermis generated keratohyalin granule-like structures that contained HRP, thereby providing definite evidence for localization of this protein in the granules [22]. Electrophoresis of proteins in the granules revealed a main fraction with a mass of 46.5 kDa. In a study of mouse epidermis, Balmain et al. [23] found that histidine-rich proteins with a range of 70 kDa to 120 kDa were rapidly synthesized in vitro, whereas a 27 kDa protein appeared simultaneously with the production of mature keratohyalin granules in fetal epidermis [24]. Interestingly, the high concentration of histidine detected by the Pauly reaction did not persist as the cells moved into the cornified layer [5,6,17]. Likewise, [^3^H]histidine that was incorporated into proteins that were synthesized in the granular layer was lost as the cells moved outward [12,19], which suggested that the protein was degraded to soluble products.

The biochemical work on keratohyalin granules that began in Bernstein’s laboratory had revealed the unusual composition of HRP, and, combined with the unconventional protocol for purification of the protein, initiated a path of research that led to characterization of a “remarkable” protein, as described by Brown and McLean [25]. Beverly Dale introduced purification of epidermal proteins by ion-exchange chromatography in 4 M urea, with which she obtained a ‘basic protein’ with properties similar to HRP [26]. Steinert, Dale and their colleagues [27] succeeded in further purifying the protein to homogeneity in 8 M urea, with final steps in dilute formic acid, from newborn mouse and rat epidermis. The mouse and rat proteins were estimated by gel electrophoresis to have masses of 30 kDa and 48 kDa, respectively. However, equilibrium sedimentation in 6 M guanidine hydrochloride yielded masses of 25.8 kDa and 38.4 kDa, respectively. The purified basic proteins, with a pI greater than pH 9.5 [26,27,28], contained about 8% histidine and 12 or 14% arginine, but little, if any, lysine, valine, leucine, tyrosine, phenylalanine or tryptophan and no cysteine or methionine. 

Peter Steinert and colleagues had extensively studied the intermediate filaments of keratin [29]. When these filaments were prepared from subunits purified from mouse, bovine and hamster tissues and mixed with HRP purified from mouse stratum corneum, a rapid increase in turbidity occurred with formation of macrofibrils. Analysis of these complexes revealed 2 mol of HRP per 3 mol of keratin intermediate filament subunits. Control experiments showed that the basic protein did not interact specifically with F-actin or tubulin. These findings were the first demonstration of the function of HRP, and Steinert, Dale, and their colleagues bestowed upon the protein the name ‘filaggrin’ (fil-ăg’-grin) for its ability to aggregate keratin intermediate filaments [27]. In contrast to filaggrin, keratin is synthesized in cells of the basal and spinous layers and becomes the bulk of the protein content in corneocytes [30].

## 3. Discovery of the Precursor, Profilaggrin

Incorporation of labeled histidine into an immunologically cross-reactive, high-molecular-mass protein (>300 kDa) revealed that filaggrin was synthesized as a large precursor and processed to smaller units [19,31]. In 1980, Lonsdale-Eccles et al. [32] reported purification of a partially processed, basic protein from the stratum corneum of newborn rats that had a pI as low as pH 6.9. The low pI resulted from 15 to 20 mol of covalently bound phosphate per mol of the protein. A precursor/product relationship between the high-molecular-weight protein and the smaller, monomeric HRP was shown by incorporation of [^3^H]histidine and [^32^P]orthophosphate into newly synthesized proteins in punch biopsies of human skin. At the end of a 1-h labeling period, a high-molecular-mass (>200 kDa) protein was labeled with both isotopes. After a 15-h chase, the 37 kDa monomer contained nearly all the [^3^H]histidine but none of the [^32^P]orthophosphate [33]. Thus, the precursor was extensively phosphorylated after synthesis in the granular layer of the epidermis, presumably to prevent premature association with, and aggregation of, keratin filaments, but as the cells differentiated into corneocytes, the protein was dephosphorylated and processed to monomers. The unique function of filaggrin in the specific aggregation of intermediate filaments led in 1985 to naming the high-molecular-mass precursor ‘profilaggrin’ [33]. The hypothesis that the gene for profilaggrin consists of tandem repeats separated by short linker sequences was demonstrated in 1986 by Haydock et al. [34] and in 1987 by Rothnagel et al. [35] from sequence analysis of cDNA clones for mouse and rat profilaggrin. 

In 1989, Steinert’s research team at the National Institutes of Health, in collaboration with Bernstein at the University of Michigan, Croce at Temple University, Parry at Massey University and Lessin at the University of Pennsylvania, published the partial structure of the human profilaggrin gene, which contained 12 repeating units [36]. The nucleotide and deduced amino acid sequences showed that each repeat contains 324 amino acids, with considerable sequence variation among the repeats. Each repeat of the human protein is separated by a conserved linker of seven amino acids with the sequence FLYQVST [36], whereas the rat and mouse profilaggrins have linkers that contain the sequence VYYY [37]. Cleavage of the protein and trimming of the linker provide filaggrin monomers with 317 amino acids and a mass of 37 kDa. Gan et al. [38] showed that the gene encoding this polyprotein in humans contains 10, 11 or 12 repeats and that different individuals may contain one or two of these three genes, among the two copies each person carries, because of allelic differences. As deduced from the cDNA sequence, the human protein contains 12% histidine, 10% arginine, 15% glycine and 25% serine [36]. The high content of serine provides ample sites for phosphorylation. In situ hybridization with antisense RNA probes exclusively decorated cells in the granular layer, which confirmed that profilaggrin mRNA is transcribed only as keratinocytes differentiate into corneocytes [35,36]. Completion of the structure of the gene was accomplished by analysis of the 5′- and 3′-regions [39,40], which revealed an S-100-like N-terminal domain that contains two EF-hand sequences that bind Ca^2+^. The full-length protein described in the National Center for Biotechnology Information contains 4061 amino acids, with a mass of 435,170 Da, which accommodates 10 filaggrin units. Figure 3 illustrates the structure of the human profilaggrin gene. 

Remarkably, the mouse [41] and rat [42] profilaggrin genes were found to contain 20 repeats. Nearly equal amounts of two different, randomly arranged repeats, 250 and 255 amino acids in length, occur in the mouse protein, and are separated by short linker sequences. The mouse and rat proteins have 60% homology between the sequences [39], but both have less sequence homology with the human protein. Nevertheless, the compositions of the proteins are similar, with the mouse protein containing 9% histidine, 12% arginine, 17% glycine and 20% serine [35]. The inclusion of an S-100-like domain in the N-terminus of profilaggrin from all of these species is interesting in light of the requirement for elevated Ca^2+^ levels for expression of the profilaggrin gene and keratinocyte differentiation [43,44,45]. Ca^2+^ binding by these domains may be largely responsible for the peak of the Ca^2+^ gradient in the granular layer [46]. Given the importance of Ca^2+^ in the process of cornification, Aho et al. [47] found that overexpression of the N-terminal domain surprisingly inhibited expression of profilaggrin and other major proteins required for keratinocyte differentiation. Complete silencing of the profilaggrin gene resulted in hyperproliferation of keratinocytes, which suggested that the S-100-like domains are required to maintain the concentration of free Ca^2+^ needed for control of these processes. 

## 4. The Short Life of Filaggrin

Upon synthesis, profilaggrin is heavily phosphorylated and is incorporated into keratohyalin granules. As cells of the granular layer move through the transitional zone into the inner layers of the cornified layer, the protein is dephosphorylated, and a complex group of proteases cleaves the precursor into monomeric filaggrin, which binds to keratin intermediate filaments and causes collapse of the cytoplasm into the predominant component of the flattened corneocytes [25,27,48]. As the cells move outward, only the first few inner layers of the cornified layer contain filaggrin, whereas the outer layers retain keratin [49]. The absence of histidine by histochemical analysis and [^3^H] from labeled histidine in the outer cornified layers revealed that filaggrin is degraded as corneocytes proceed through the inner layers of the stratum corneum. 

A flowchart that illustrates the steps in this process is provided in Figure 4 (see legend for reference numbers). The N-terminal A domain, which contains the Ca^2+^-binding EF-hand sequences, is cleaved from profilaggrin prior to processing of the remainder of the protein. Skin-specific retrovirus-like aspartic protease (SASPase) cleaves the linker sequences to yield active monomers that bind to keratin intermediate filaments. Knock-out of the serine proteases CAP1/Prss8 and matriptase/MT-SP1 caused incomplete processing of profilaggrin and little, if any, monomeric filaggrin, which suggested that these proteases are also involved in processing profilaggrin to the monomeric form. Upon completing its function in facilitating collapse of keratin filaments, filaggrin is further degraded by calpain 1, caspase 14 and bleomycin hydrolase to free amino acids (Figure 4B). The final mixture of amino acids, along with urocanic acid (UCA) produced by deamination of histidine, 2-pyrrolidone carboxylic acid (PCA) produced by cyclization of glutamine and other metabolic products, have been designated the ‘natural moisturizing factor (NMF)’ and aid in maintaining hydration of the stratum corneum. The activity of most of these proteases is Ca^2+^-dependent.

## 5. Mutations That Cause Loss of Filaggrin

A major discovery published in 2006 revealed the cause of the complete loss of filaggrin in ichthyosis vulgaris and atopic dermatitis (AD), diseases that result from defective formation of the surface barrier [64]. The loosened cornified layer of the skin in these cases allows water to escape and allergens to gain access to internal cells [65]. Irwin McLean’s team at the University of Dundee identified mutations in the first repeat in exon 3 of the profilaggrin gene (R501X generates a nonsense stop-codon, and the 2282del4 deletion also results in a stop-codon), thus revealing the genetic cause of these diseases [25,64,65,66]. The null mutations R501X and 2282del4 are prevalent in European and Asian populations and lead to the complete absence of profilaggrin in homozygous individuals. Within the next 5 years, an extensive mutation map was generated that showed significant differences between European and Asian populations (Figure 5). A recent analysis of 126 patients with atopic dermatologic disorders in Saudi Arabia detected 227 variants, including missense, silent, nonsense, frameshift and noncoding mutations in exon 3 of the profilaggrin gene [67]. Within the decade following the publication by McLean’s team in 2006, a nearly a five-fold increase in the number of research articles related to filaggrin appeared in the literature (Figure 6). Several excellent reviews describe the research productivity during this decade [25,68,69,70]. 

Mutations in the gene encoding profilaggrin are the strongest risk factors for skin diseases. About 50% of AD patients carry loss-of-function mutations in filaggrin [71]. Esparza-Gordillo et al. [72] indicated that 10% to 20% of people in industrialized countries suffer from AD (eczema), with a strong disposition in children when the mother carries an *FLG* mutation. Ichthyosis vulgaris is one of the most common skin diseases, characterized by dry, flaky skin, with a prevalence of at least 1 case per 250 persons. The complexity of these diseases was expanded by Butler et al. [73], who identified twelve distinct diseases within the spectrum of AD. AD is an inflammatory condition, often with open lesions, that affects 14% of children in the US [74] and up to 25% in the UK and Scandinavia [75]. AD is the most common chronic skin disease in early childhood and peaks at 2 to 3 years of age, after which it declines to a few percent in adults [75,76]. 

McAleer and colleagues measured the amounts of natural moisturizing factor (NMF) and other degradation products of filaggrin in the skin of *FLG^+/+^* children as indicators of filaggrin synthesis [76]. The nearly five-fold increase in NMF and a 15-fold increase in free histidine on the cheeks of children during the first 3 years of life indicated that the rate of synthesis of this short-lived protein is quite low after birth, and maturation of the epidermis develops slowly. The immaturity of the skin during this period reveals vulnerability for sensitization to food allergens and other life-long allergies. 

Interestingly, defects in processing profilaggrin appear to be more severe than a complete lack of filaggrin. Mice with a deficiency in the protease CAP1 [53] processed profilaggrin to short oligomers but no further, which resulted in an impaired surface barrier, rapid dehydration and death shortly after birth. Pleotropic effects of a deficiency of matriptase/Mt-SP1 also were related to a defect in processing of profilaggrin during cornification of the epidermis, but the minimal processing in these animals seemed to cause less severe disease [54].

A deficiency in filaggrin is not the only cause of AD. Multiple mutations in other genes [71] and also environmental factors [77,78,79] may cause AD. In fact, the majority of cases of AD have environmental causative or comorbidity factors. Nevertheless, the deficiency of filaggrin has been an instructive case study. Thyssen and Kezic [80] provided an excellent overview of the multifactorial environmental and genetic causes of AD. An extensive review by Dębińska [81] describes treatments for AD currently approved or in clinical trials, with specific focus on factors that regulate expression of the filaggrin gene.

Several animal models of AD have been developed [82,83]. Stout et al. [84] developed an interesting approach to treatment of filaggrin deficiency in the ‘flaky tail’ strain of mice by linking a recombinant filaggrin monomer to a cell-penetrating peptide derived from the HIV TAT protein. After topical application, the 50 kDa fusion product was processed to a 28 kDa protein, which corresponded to the normally processed filaggrin monomer and restored normal function to mouse skin. Whereas the ‘flaky tail’ strain is deficient in filaggrin, the protein is not completely absent. Kawasaki et al. [85] generated a homologous recombination vector that eliminated the promoter sequences for mRNA transcription and all ATG start codons for synthesis of this protein, which resulted in complete loss of profilaggrin and a phenotype similar to AD in humans. The condition has also been simulated experimentally by applying a solution of SDS to the skin [86]. With humans, a 24-h treatment with 1% SDS caused expression of profilaggrin to decrease dramatically but then increase over controls after 4 days post-treatment [87]. Application of 17% SDS for 7 h caused the epidermis to become leaky, after which restoration of the surface barrier was studied by measurement of the transepidermal water loss (TEWL) during treatment with ceramide mixtures [88]. 

Healthy skin requires continual loss of the outer layers by corneocyte desquamation; the outer layers are replaced by dividing cells in the basal layer. However, an excessive loss of corneocytes causes an extreme type of lesional disease similar to AD, known as Netherton syndrome (NS), which results in inflammation and allergic reactions [89,90]. NS is a life-threatening disorder that affects approximately one in 200,000 newborn children [89]. This disease is mediated by kallikrein-related serine peptidases (KLKs), whose normal function is to degrade corneodesmosomes and allow detachment of superficial corneocytes. KLK5 and KLK7 are synthesized as zymogens and activated by cleavage of the precursor forms by mesotrypsin and matriptase [91]. Excessive loss of corneocytes by overactive KLK5 and KLK7 is controlled by the serine protease inhibitor lymphoepithelial Kazai-type-related inhibitor (LEKT1) encoded by the *SPINK5* gene. LEKT1 is also degraded by mesotrypsin, which is activated by enteropeptidase in the granular layer [91]. NS is caused by a loss-of-function mutation in the *SPINK5* gene, which allows heightened activity of KLKs. Knock-out of KLK5 reversed NS symptoms in a SPINK5-deficient mouse model [89,90,91,92]. Because of lethality of the *Spink5^−/−^* knock-out, immunological consequences of the mutation were studied in skin grafts on nude mice. Activation of protease-activated receptor 2 (PAR2), a major effector of the inflammatory response, by KLK5 led to recruitment of eosinophilic and mast cells and formation of lesions [92]. 

Early histological studies found that keratohyalin granules were not present in the epidermis in psoriatic lesions [6,18], and HRP was absent in extracts of these lesions. However, the protein and granules were present in adjacent ‘normal’ skin [18,19]. Although symptoms of psoriasis are similar to those of AD, mutations in filaggrin are not causative for psoriasis [93,94].

## 6. Keratinocyte to Corneocyte Transition

The context in which filaggrin functions is the transition of keratinocytes to corneocytes. More specifically, this transition involves formation of the cornified envelope and the conversion of cell cytoplasm to an insoluble matrix. Proteins synthesized in the granular layer are organized into a structure, referred to as the ‘cornified envelope,’ that lines the cytoplasmic surface of the cell membrane [95,96,97,98]. Transglutaminases catalyze cross-links between these proteins, which are essential for stabilization of the complex. Three isozymes of the Ca^2+^-dependent transglutaminase family, TGM1, TGM3 and TGM5, are expressed in the epidermis and function within terminally differentiating keratinocytes [98]. An early step in formation of the cornified envelope is attachment of involucrin to the inner surface of the cell membrane by cross-links catalyzed by the essential enzyme TGM1 [96,98]. Although involucrin contains 39 repeats of a 10-amino-acid sequence, with each containing three glutamine (Q) residues, only one in this 585-amino acid protein—within the sequence ELPEQ**Q**VGQP (the reactive Q is underlined and bold)—is a substrate for TGM1 in vitro unless the protein is proteolytically degraded [99]. Loricrin, an abundant protein that comprises about 80% of the cornified envelope, is synthesized in cells of the granular layer and forms oligomers with small, proline-rich proteins [100,101]. These complexes are stabilized by TGM3 and then fixed onto the involucrin-containing scaffold by formation of cross-links catalyzed by TGM1 [95]. The keratin filament-filaggrin complex and several minor proteins, such as elafin, are added sequentially to the inner surface of the cell membrane. Keratin and a small amount of filaggrin are cross-linked to loricrin but not to each other [101]. Cross-links, mostly to loricrin, stabilize the cornified envelope and generate the insoluble matrix of corneocytes. Mutations in the TGM1 gene cause the skin disorder lamellar ichthyosis, and *TGM1^−/−^* knockout mice die within a few hours after birth [96,102]

Liedén et al. [103] and Su et al. [104] demonstrated a remarkable increase in expression of these transglutaminases in the epidermis of patients with AD. Whereas TGM1 and TGM3 are intracellular enzymes, repair of damage to the epidermis appears to also require extracellular transglutaminase activity. TGM2 is ubiquitously expressed and, unlike other members of the family, is found in the extracellular space [105,106,107,108]. Expression of TGM2 is induced in inflamed and wounded tissues [105,109,110,111,112] and also by dexamethasone [113]. TGM2 normally occurs in an inactive, closed conformation but is rapidly converted to the open, active form by injury [109,114]. Closure of wounds is significantly impaired in TGM2 knock-out mice [115], and processes required for wound healing and extracellular matrix remodeling are dramatically reduced when fibroblasts and macrophages are rendered TGM2-deficient by transfection with anti-sense RNA [109,110]. 

## 7. Another Role for Transglutaminase?

Whereas the corneocytes are the “bricks” of the surface barrier, ceramides and other lipids are extruded into the extracellular space to form the “mortar” [116,117]. Impermeability to water is provided by the extracellular lipids, as demonstrated by complete abrogation of the permeability barrier by the extraction of lipids [118,119]. Points of contact between corneocytes occur at villus-like structures, which at their tip contain corneodesmosomes, a complex consisting of cohesion proteins corneodesmosin, desmoglein 1 and desmocollin 1 [120,121]. Corneodesmosin is synthesized in the lower granular layer of the epidermis and is secreted from keratinocytes in vesicles as a glycoprotein [122,123]. In healthy skin, corneocytes are tightly stacked in the stratum corneum and express these villus-like structures mostly on their outermost rims (Figure 7B). Riethmüller et al. [124] discovered that these structures are approximately five-fold more numerous on homozygous *FLG^−/−^* cells than on wild-type cells and cover the entire surface of the loosely packed mutant corneocytes (Figure 7A). The less tightly stacked stratum corneum of *FLG^−/−^* individuals is also characterized by a nearly two-fold increase in TEWL [124]. Although important for stability of the stratum corneum, corneodesmosomes do not inhibit water flow between cells. However, TEWL could possibly be reduced if adherence of corneocytes in filaggrin-deficient patients was increased. 

The greater abundance of corneodesmosomes in filaggrin-deficient patients may be an attempt to compensate for the deficiency in filaggrin. Serine/glycine-rich loops in the N-terminal domain of corneodesmosin are exposed on the surface of the stratum corneum and are binding sites for colonization by *Staphylococcus aureus* and other pathogens in AD lesions [125]. Several lysine residues lie within the sequence near these loops and could serve as substrates for TGM2. An enzyme-bound thioester with the sidechain carboxyl group of a glutamate residue, formed in the first step of a cross-linking reaction, is attacked in the second step by an ε-amino group of a lysine residue in a protein to form ε(γ-glutamyl)lysine isopeptide bonds. Seemingly any lysine ε-amino group will enter the second step, but the enzyme is very specific for the glutamine residue, even when the protein is rich in glutamine. The sequences glutamine-X (any amino acid)-proline (QXP) and QXXQ are preferred substrates for TGM2, whereas QP or QXXP are not favorable substrates for the enzyme [126,127]. The paucity of accessible glutamine residues may minimize the number of cross-links between corneocytes. To overcome this restriction, a glutamine-containing, multivalent peptide, svL4 [128], which is a substrate for TGM2 (unpublished data), could provide ‘lysine-peptide-lysine’ cross-links between these corneocytes and thereby potentially tighten the surface barrier, as proposed in Figure 8. 

This approach could possibly be an effective treatment of necrotic lesions in severe cases of AD and for the repair of wounds. Additional cross-links between corneocytes may also resist excessive desquamation. Restoration of the surface barrier would allow the epidermal Ca^2+^ gradient to reform, which is critical for keratinocyte differentiation and healthy skin [43,44,45,46,105,129]. 

This brief review provides an account of the intensive path of research, which in a relatively short period of time since its humble beginnings has achieved a level of understanding of cornification of the skin that is making a significant impact on quality of life. 

## Figures and Tables

**Figure 1 ijms-23-01455-f001:**
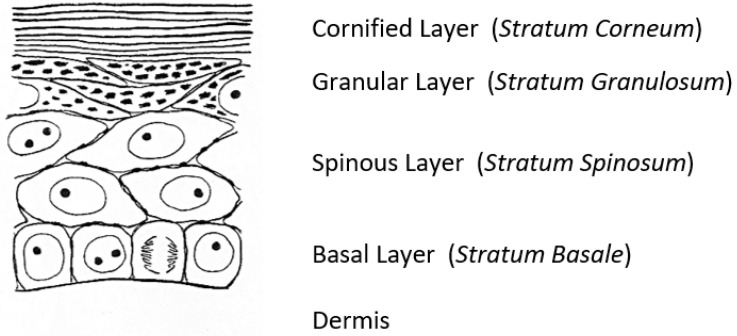
A sketch of the epidermis of the newborn rat [4].

**Figure 2 ijms-23-01455-f002:**
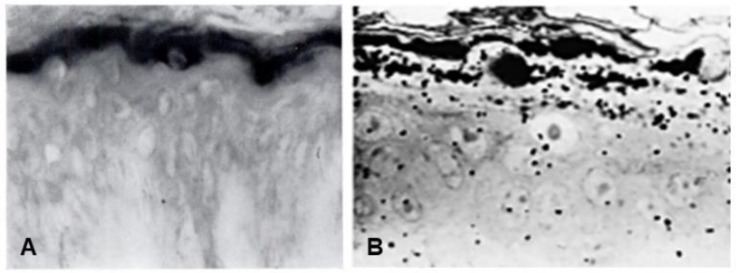
(**A**) Human skin stained with the Pauly reagent for histidine [5]. (**B**) Light microscopic radioautograph of newborn rat epidermis 15 min after intraperitoneal injection of [^3^H]histidine [12].

**Figure 3 ijms-23-01455-f003:**
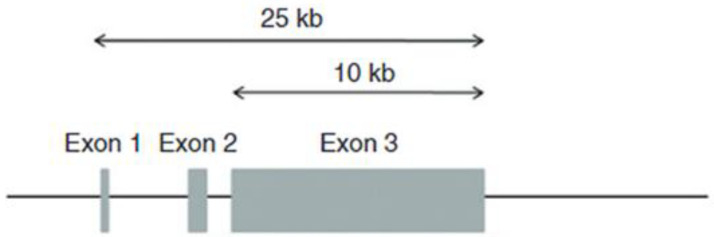
The human profilaggrin gene on chromosome 1q21. Exon 2 contains the translation initiation codon. The entire coding region of the protein is within the uninterrupted exon 3.

**Figure 4 ijms-23-01455-f004:**
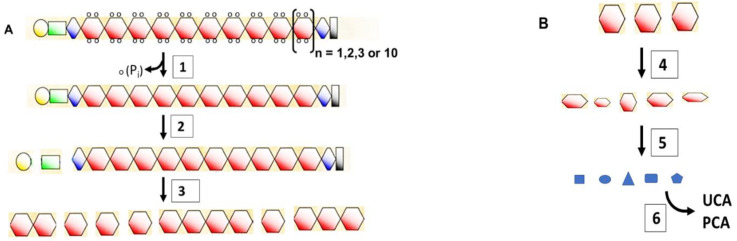
Flowchart of the processing of profilaggrin to free amino acids. (**A**). Proflaggrin to filaggrin. (1) Phospho-profilaggrin is dephosphorylated by phosphatases. (2) The A (yellow) and B (green) domains are cleaved from profilaggrin by furin, PACE4 (Paired basic Amino acid Cleaving Enzyme 4) and endoproteinase-1 (PEP1) [50,51]. (3) Linker sequences of human profilaggrin (FLYQVST) are cleaved by skin-specific retrovirus-like aspartic protease (SASPase) [52], channel-activating serine protease (CAP1) [53] and matriptase (MT-SP1) [54] to monomeric filaggrin. Aminopeptidases and carboxypeptidases are likely involved in trimming the new termini [37]. (**B**). Filaggrin to NMF. (4) Arginine residues in filaggrin and keratin are converted to citrulline by peptidylarginine deiminase (PAD1 or 3) [49,55]. Deiminated filaggrin is cleaved by calpain-1 and caspase-14 (at VSQD and HSED sequences) to filaggrin fragments [55,56,57]. (5) Filaggrin fragments are digested by neutral cysteine protease (bleomycin hydrolase) to amino acids [57]. Aminopeptidases and carboxypeptidases are also likely involved. (6) Histidine is converted by histidine deaminase to *trans*-urocanic acid (UCA), which has a UV spectrum similar to that of nucleic acids and proteins [58] and provides a natural sunscreen [59]. Glutamine is non-enzymatically converted to 2-pyrrolidone carboxylic acid (PCA). These hydrophilic final products contribute to the moisturizing of the skin [56,57,60,61]. Kezic et al. [62] and de Veer et al. [63] provided excellent reviews of proteolytic processing of filaggrin and differentiation of corneocytes.

**Figure 5 ijms-23-01455-f005:**
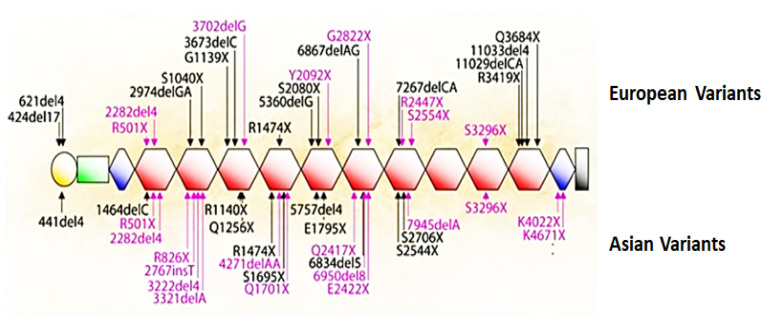
Filaggrin mutations among populations. Mutations that are recurrent in European and Asian populations are indicated in red, and rare or family specific mutations are in black. These mutations are either nonsense mutations or out-of-frame insertions or deletions that are predicted to cause loss-of-function. Exon 3 of the gene contains the complete sequence for profilaggrin, shown here with 10 repeats as orange hexagons. The yellow circle is the S-100-like Ca^2+^-binding domain, the green rectangle is the B domain, and the blue diamonds are incomplete repeats [65] (with permission from A. D. Irvine).

**Figure 6 ijms-23-01455-f006:**
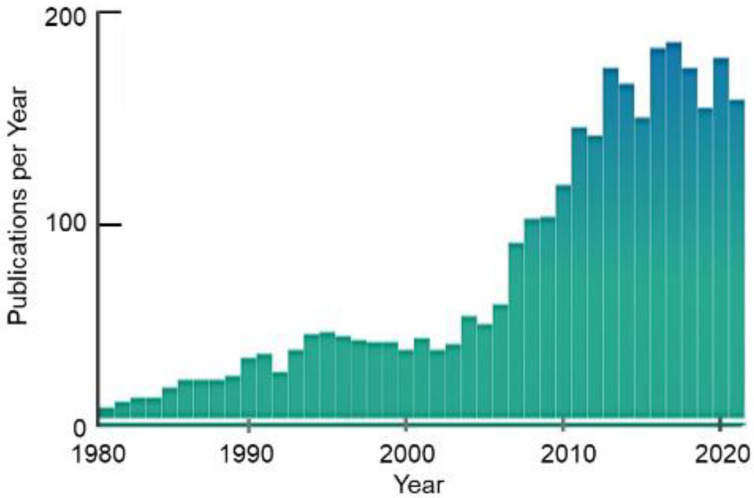
Number of publications that refer to filaggrin as a function of year.

**Figure 7 ijms-23-01455-f007:**
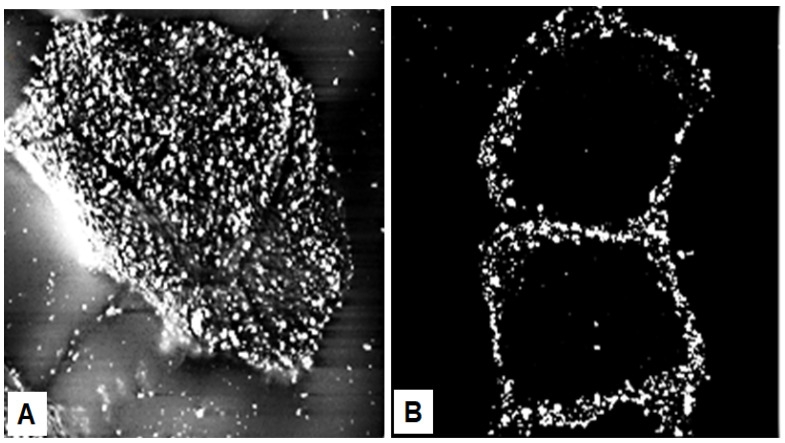
Corneodesmosomes. (**A**) Filaggrin-deficient (*FLG^−/−^*) corneocytes from tape-stripped stratum corneum of patients with AD express corneodesmosin on villus-like projections that cover the entire cell surface. (**B**) In contrast, corneocytes from the tightly packed stratum corneum of wild-type (*FLG^+/+^*) patients contain corneodesmosin mostly on lateral rims of the cells. The images were prepared by incubating corneocytes with anti-corneodesmosin antibodies followed by immunogold labeling and scanning electron microscopy [124] (with permission from S. Kezic).

**Figure 8 ijms-23-01455-f008:**
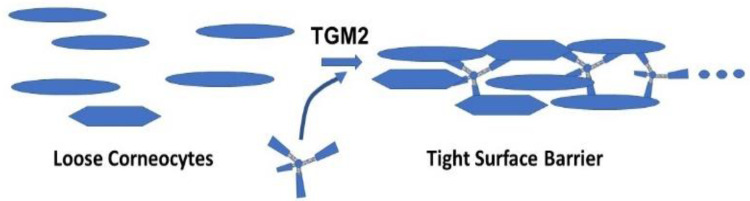
Model for the role of a tetravalent peptide in facilitating cross-linking of corneocytes by TGM2.

## Data Availability

Not applicable.

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
