# Peer review of "The Discovery and Function of Filaggrin"

_ijms, 2022, doi:10.3390/ijms23031455_

Round 1

Reviewer 1 Report

The present article is a brief review on the history and biology of filaggrin. The review is well-written with important historical information.

I only have few minor suggestions.

  1. In the section of AD models the authors should cite the Spink5-/- as a model for AD (Briot et al. 2009 Kallikrein 5 induces atopic dermatitis-like lesions through PAR2-mediated thymic stromal lymphopoietin expression in Netherto syndrome. J Exp Med 206: 1135-1147) and potentially the identification of KLK5 as a target for related diseases (Furio et al. 2015 KLK5 inactivation reverses cutaneous hallmarks of Netherton syndrome. PLoS Genet 11: e1005389).
  2. I suggest a small discussion on the proteolytic processing of filaggrin (de Veer et al. 2014 Proteases: common culprits in human skin disorders. Trends Mol Med 20: 166-178.)

Author Response

Response to Reviewer 1:

We appreciate your excellent comments and have revised the manuscript accordingly.  A paragraph was added to the section on mouse models to include information on the Netherton syndrome with specific reference to the loss-of-function mutations in the Spink5 gene (lines 338 to 356 in the revised version).  In regard to the second comment, the section on processing of profilaggrin to monomeric filaggrin and the subsequent degradation to amino acids was significantly expanded (lines 210 to 256) and included the specific proteolytic activities involved.  These processes are illustrated in the additional Figure 4.   

Reviewer 2 Report

 I peer-reviewed the manuscript entitled  "Filaggrin".

The authors presented a paper without a clearly defined topic, which is reflected also in the title of this work. It is very difficult to comprehensively summarize decades of research on filaggrin on 10 pages. Therefore I propose that the authors focused their research on selected aspects, either summarizing it's history or clinical relevance in dermatology.

That being said, the manuscript is hard to follow, as it is highly interdisciplinary, but lacks proper backgroud. It mostly lists facts that seem not to be related to one another. It took me several times to understand the relationship between loosly listed sentences to form a coherent point. 

I also would like to point out the fact that the authors do not provide an aim of their manuscript at the beginning, and therefore this manuscript reads more like a textbook chapter than a research paper. 

Nevertheless, I think a good review of filaggrin function (especially in clinical dermatology) with a background on immunity, is necessary. 

I also feel that the paper would greatly benefit if some of the paragraphs (especially biochemical processes) were presented graphically, to make it easier to understand. 

 More detailed corrections are presented in the pdf attached below. 

Author Response

Response to Reviewer 2:

We very much appreciate the excellent and constructive comments and have extensively revised the manuscript.  Our purpose, and thus the emphasis, was to describe the history of the work that went into an understanding of the function of this remarkable but short-lived protein.  Specifically, the title was changed as recommended and the initial sentences were revised to indicate the purpose of the article.  The section on processing of profilaggrin to filaggrin and the subsequent degradation to amino acids (lines 210 to 256) was expanded and.  as suggested, Figure 4 was created to graphically illustrate the proteolytic activities.  The suggestions marked on the manuscript were noted in our revisions by reordering sections, expanding, and clarifying.  As a result, the manuscript is eminently improved.  Not being clinicians, we focused instead on the biochemical and genetic developments.   Significant discussion was given to mutations that lead to diseases (but without discussion of clinical symptoms) and some of the animal models that allow research on these diseases.  The final section provides the overall context in which filaggrin is required to form the functional surface barrier and the related role of essential transglutaminases in this process.